# Diversity, Composition, and Ecological Function of Endophytic Fungal Communities Associated with *Erigeron breviscapus* in China

**DOI:** 10.3390/microorganisms13051080

**Published:** 2025-05-06

**Authors:** Yi Zhao, Hui Wu, Fang Wang, Liangzhou Zhao, Weijun Gong, Haiyan Li

**Affiliations:** 1Key Laboratory of Chemistry in Ethnic Medicinal Resources, Yunnan Minzu University, Kunming 650500, China; febflower-zy@163.com (Y.Z.); wangfang@ymu.edu.cn (F.W.); 2Medical School, Shanghai Jiao Tong University, Shanghai 200025, China; hwu@shsmu.edu.cn; 3Medical School, Kunming University of Science and Technology, Kunming 650500, China; zhaoliangzhou2021@163.com (L.Z.); weijung98@126.com (W.G.)

**Keywords:** endophytic fungi, *Erigeron breviscapus*, diversity, ecological function, high-throughput sequencing

## Abstract

Endophytic fungi (EF) play crucial roles in accelerating the accumulation of medicinal components and regulating the survival and reproduction of medicinal plants. *Erigeron breviscapus* is a well-known traditional Chinese medicinal plant with effective therapeutic effects and a wide application, but little is known about the diversity, community structure, and ecological roles of EF in this plant. Here, the EF communities associated with different tissues of *E. breviscapus* at two seasons were studied by high-throughput sequencing methods. Furthermore, FUNGuild was performed to predict the ecological functions of the fungi. *Didymella* was found to be the most dominant genus across all four tissues, followed by *Plectosphaerella*, *Filobasidium*, *Cystofilobasidium*, and *Cladosporium*. Notably, the dominant and unique genera and biomarkers of four tissues were different. Interestingly, it was found that the roots had the highest fungal richness and diversity in summer. Moreover, both PCoA plots and PERMANOVA analyses indicated that the tissue and season were main factors contributing to the differences in the fungal communities of *E. breviscapus*. FUNGuild prediction revealed that pathotroph–saprotroph fungi and undefined taxa accounted for a high proportion in the EF of *E. breviscapus*. We also found some valuable endophytes that encouraged deeper investigation. These findings provide a theoretical reference for the further development and utilization of EF resources in *E. breviscapus*.

## 1. Introduction

*Erigeron breviscapus* (Vant.) Hand-Mazz. has been used as an herbal medicine in China for more than 1000 years [1]. *E. breviscapus* extracts have been extensively utilized for treating cerebral ischemia, cerebral infarction, cardiovascular disease, pulmonary heart disease, diabetes complications, and nervous system diseases [2,3]. Recent studies have confirmed that *E. breviscapus* might provide a promising prospect for Alzheimer’s disease (AD) treatment [4]. Currently, about 4000 tons of *E. breviscapus* materials are collected and used in the breviscapine industry per year in China [5], thereby greatly depleting *E. breviscapus* germplasm resources.

*E. breviscapus* is endemic to southwestern China [6,7]. This species is mainly distributed in the Yunnan, Guizhou, and Sichuan provinces of China, and it grows in hillsides, grasslands, and open woodlands within an altitude of 1000–3500 m [8]. *E. breviscapus* is an herbaceous and perennial flowering plant that belongs to the Asteraceae family [9]. Its flower heads have yellow disk florets and showy blue to purple rays. Traditionally, whole grass can be used as medicine and harvested many times each year. However, the plant supply of *E. breviscapus* is still insufficient in the market [1,10].

Endophytes are thought to be extremely important plant partners [11]. They live asymptomatically in plant tissues for all or a certain period of life [12]. Host plants usually provide a multitude of niches for endophytic fungi. In turn, endophytes confer on their hosts beneficial effects such as resistance to various biotic and abiotic stresses, including salinity, drought, pathogens and metal toxicity, etc. [13]. Importantly, endophytic fungi are often embedded in their host’s metabolic networks. They may biosynthesize the same metabolites as the host or produce distinct host plant metabolites [14,15]. Some of these fungal metabolites can also exhibit medicinal activities, demonstrating potential medicinal value. Moreover, recent studies in several different experimental systems have demonstrated that endophytes can promote bioactive constituent accumulation in medicinal plants [16,17]. Therefore, characterizing the composition and diversity of fungal endophytes may provide a new way to discover novel medicinal compounds and understand the role of fungal species in ecosystems.

Previous studies have demonstrated that host species, plant tissues, and seasons can affect fungal community composition [18,19]. However, the effect of these factors on the endophytic fungal community of *E. breviscapus* is still unknown. Thus, the present study was designed to examine the fungal endophytic community of *E. breviscapus*’s different tissues at two contrasting seasons through high-throughput sequencing technology.

## 2. Materials and Methods

### 2.1. Sample Collection

Plants of *E. breviscapus* in the flowering period were collected from Luxi county, Honghe autonomous prefecture, Yunnan Province, Southwest China (24°40′25′′ N, 103°46′24′′ E, 1829 m asl) in January and July 2022. On the day of sampling, the average temperatures were 15.6 °C (January) and 21.8 °C (July). The area is characterized by a northern subtropical monsoon climate. The soil pH of the sampling location in this study was about 6.3. *E. breviscapus* samples were taken from an experimental plot. Sampling and further analysis were performed in triplicate. At each sampling time, a total of 40 healthy plants (from 4 sampling points, 10 plants for each sampling point, and a sampling distance of 10 m between each point) were collected. Samples from three randomly chosen sampling points were analyzed, while the remaining samples were reserved as spares. All samples were placed in sterile plastic bags, marked, and brought back to the laboratory and processed within 24 h.

### 2.2. Sample Pretreatment

Fresh plants were washed in running tap water to remove soil and other contaminants. These plants were separated as flowers (F), roots (R), stems (S), and leaves (L). We mixed the same organs from 10 plants at each sampling point into one sample, resulting in three individual biological replicate samples from three sampling points. Each tissue type contained triplicate biological replicates. We acquired a total of 24 samples (F-1-1, F-1-2, F-1-3, R-1-1, R-1-2, R-1-3, S-1-1, S-1-2, S-1-3, L-1-1, L-1-2, L-1-3, F-7-1, F-7-2, F-7-3, R-7-1, R-7-2, R-7-3, S-7-1, S-7-2, S-7-3, L-7-1, L-7-2, L-7-3) for subsequent analysis. Then, the surface sterilization was performed as follows: the samples were further cut into segments and immersed in 75% ethanol for 3 min, rinsed 3 times with sterile water, followed by 5% sodium hypochlorite for 2 min and finally washed 3–5 times with sterile water [20]. Surface-sterilized tissues were dried on sterilized filter paper. To confirm the sterilization efficiency, the final water rinse was plated onto a potato dextrose agar (PDA) plate, and if no colonies were observed, the sterilization was considered successful [21].

### 2.3. DNA Extraction and Sequencing

The surface-sterilized plants were homogenized in sterile mortars with liquid nitrogen. Afterwards, the total genomic DNA was extracted using a HiPure Soil DNA Mini Kit (Magen, Guangzhou, China) and was verified by gel electrophoresis (0.8% agarose). For the amplification of the fungal ITS1 gene region, primer ITS1F (5’-CTTGGTCATTTAGAGGAAGTAA-3’) and ITS2 (5’-GCTGCGTTCTTCATCGATGC-3’) were used [22]. PCR reactions contained the following: Q5 High-Fidelity 2× Master Mix 12.5 µL (New England Biolabs, Ipswich, MA, USA), 1.25 µL primer (10 µM each), dNTP mixture (200 μM), template DNA (30 ng), and ddH_2_O in a volume of 25 µL. The PCR program consisted of an initial denaturation step of 95 °C for 5 min, followed by 33 cycles of denaturation at 95 °Cfor 1 min, annealing at 60 °C for 1 min, an extension at 72 °C for 1 min, and a final extension of 7 min at 72 °C. The PCR products were recovered by 2% agarose gel and purified by AxyPrep DNA Gel Extraction Kit (Axygen Biosciences, Union City, CA, USA). Purified DNA was quantified using the ABI StepOnePlus Real-Time PCR System (Life Technologies, Foster City, CA, USA). Paired-end sequencing (2 × 250 bp) was carried out on an Illumina Novaseq 6000 sequencing platform at Guangzhou Gene Denovo biological technology company (Guangzhou, China). The Illumina sequencing data obtained in these experiments are publicly available in the NCBI Sequence Read Archive under project numbers PRJNA 901262 and PRJNA 901633.

### 2.4. Processing of Sequencing Data

The original sequence was quality controlled and filtered by FASTP (version 0.18.0) [23]. Paired-end reads from the original DNA fragments theoretically were merged as raw tags using FLASH (version 1.2.11) [24]. All sequences were denoised, as well as trimmed, for barcodes and primers. Chimeric sequences, as well as chloroplast and mitochondrial sequences, were removed using UCHIME [25]. The clean tags were clustered into operational taxonomic units (OTUs) based on a 97% similarity criterion using UPARSE (version 9.2.64) [26]. The tag sequence with highest abundance was selected as a representative sequence for further annotation. Taxonomy was assigned using the UNITE database (version 8.3) [27].

### 2.5. Statistical Analysis

The alpha diversity indexes including Shannon, Simpson, Chao1, and ACE index were calculated by QIIME (version 1.9.1) [28]. The differences in alpha diversity indexes among four tissues were analyzed by a one-way ANOVA (response variable residuals are normally distributed and variances of populations are equal) or Kruskal–Wallis test in SPSS (version 29.0). Venn diagrams were plotted with the package “Venn Diagram (version 1.6.16) [29]”. Principal coordinate analysis (PCoA) based on Unweighted UniFrac distance were used to investigate microbial community dissimilarity [30]. Permutational ANOVAs (PERMANOVAs) were performed with the function “adonis” in the vegan package (version 2.5.3) of R [31]. FUNGuild (version 1.0) was employed to predict the ecological functions of the fungi [32]. LEfSe (linear discriminant analysis effect size) analysis was implemented on the Galaxy platform (version 1.0), with a threshold score set to 4.0 [33].

## 3. Results

### 3.1. Sequencing Yields

We analyzed endophytic fungal communities associated with 24 samples taken from various parts of *E. breviscapus* in two different seasons. A total of 3,087,176 raw reads were obtained from the sequencer, and 2,575,479 tags were produced after splicing paired-end reads. The raw tags passed through quality control to yield clean tags, and then chimera filtering was performed to produce effective tags for further analysis. The read lengths ranged from 201 to 474 bp. The Q20 values were greater than 90.49%, and the GC (GC base) content values were 43.90–48.71% (Appendix A).

Rank abundance curves can intuitively reflect both the abundance and evenness of species [34]. The curves of root and stem samples display a wider and gentle distribution compared to those of leaves and flowers. This observation indicated that the endophytes of roots and stems were relatively greater and more abundant than that of leaves and flowers (Figure 1). The rarefaction curves of all the plant samples tended to approach the saturation plateau (Appendix A). The results indicated that the sequencing depth and the number of OTUs were sufficient.

### 3.2. Fungal Endophytic Community Composition

In total, 811 OTUs were obtained from 24 samples of *E. breviscapus*. Among them, 540 and 780 different OTUs were detected from the January and July samples, respectively. The Venn diagram (Figure 2a) illustrates the composition of shared and unique OTUs in different organs of *E. breviscapus*. A total of 105 fungal OTUs (12.95%) were shared in all the samples, while 38, 34, 293, and 94 OTUs were only detected in flowers, leaves, roots, and stems. Root tissues contained higher numbers of unique endophyte OTUs than other tissues.

The OTUs were assigned into 10 phyla, 30 classes, 73 orders, 153 families, 259 genera, and 261 species. It was found that fungal communities associated with *E. breviscapus* comprised ten known fungal phyla as follows: *Ascomycota* (average relative abundance 62.50%), *Basidiomycota* (23.64%), *Glomeromycota* (0.46%), *Mortierellomycota* (0.37%), *Olpidiomycota* (0.05%), *Chytridiomycota* (0.046%), *Aphelidiomycota* (0.01%), *Mucoromycota* (<0.01%), *Zoopagomycota* (<0.01%), and *Kickxellomycota* (<0.01%). Among them, *Ascomycota* and *Basidiomycota* were the dominant phyla across all the samples (Figure 3, Appendix A). However, fungal endophytic communities of *E. breviscapus* showed significant differences among different organs (Figure 3a,c). In general, *Ascomycota* was dominant in the root (74.67%) and leaf (73.52%) of *E. breviscapus*, while *Basidiomycota* was more abundant in the flower (39.59%) and stem (33.70%) samples than other samples (Figure 3a).

At the genus level (Figure 3c, Appendix A), *Didymella* was the dominant genus, existing in all four organs. Other dominant genera were *Plectosphaerella*, *Filobasidium*, *Cystofilobasidium*, etc. However, several significant differences were observed. The Venn diagram (Figure 2b) illustrates the composition of shared and unique species shared by different organs. In total, 108 different genera were identified in flower samples with 13 unique genera, of which *Didymella* (20.35%) had the highest abundance, closely followed by *Filobasidium* (17.89%) and *Cystofilobasidium* (14.66%). Conversely, 94 genera were identified in the leaf samples with 8 unique genera, of which *Didymella* (37.17%) had the highest abundance, followed by *Plectosphaerella* (19.04%). In root tissue, 133 genera were detected with 41 unique genera, of which unclassified endophytes (42.26%), *Plectosphaerella* (21.73%), and *Didymella* (15.06%) were the dominant genera. In stem tissue, 127 genera were detected, 20 of which were unique. It was found that stems were dominated by *Filobasidium* (23.25%), followed by *Didymella* (23.10%).

In addition, we observed obvious changes in the community structure with the seasons (Figure 2c,d and Figure 3b,d). The Venn diagram (Figure 2c) shows the numbers of shared and unique OTUs in different seasons. In total, 147 OTUs were shared between summer and winter. The number of unique OTUs in summer was higher than that in winter. At the phylum level (Figure 3b), *Ascomycota* (72.48%) was highly abundant in summer, while *Basidiomycota* (34.10%) was higher in winter. At the genus level (Figure 3d), *Didymella* (26.10%) and *Plectosphaerella* (16.77%) were more abundant in summer, while *Filobasidium* (17.51%) and *Cystofilobasidium* (9.14%) were mainly distributed in winter.

### 3.3. Diversity of Fungal Endophytes

Computational analyses of α-diversity estimated the richness and diversity of endophytic fungi associated with four plant tissues at OTU cutoffs of 0.03 distance units (Figure 3). The Chao1 and ACE indexes commonly reflect species richness, while the Shannon (*H*′) and Simpson indexes mainly reflect community diversity [35]. It was found that the highest index of *H*′ was observed in the root samples at two different collection times (Kruskal–Wallis test, *p* < 0.05 in July) (Figure 4a, Appendix A). Similarly, the Simpson indexes of root-associated fungi were higher compared with other tissues (one-way ANOVA, *p* < 0.05 in July) (Figure 4b, Appendix A). Remarkably, the fungal Chao1 (Figure 4c) and ACE (Figure 4d) indexes of roots were also significantly higher than other tissues (Appendix A and Appendix A), indicating that the diversity and richness of root fungal communities of *E. breviscapus* were higher than those of the other samples. That means the diversity of endophytes of *E. breviscapus* varies widely in different organs.

Interestingly, we observed a seasonal disparity in the diversity of endophytic fungi from *E. breviscapus*. Compared with winter (January) samples, the highest value of the α-diversity index was generally found in summer (July) (Figure 4). For example, in January, the highest *H*′ index was 3.43, while in July, the highest *H*′ index was recorded in roots with 5.67. The *H*′ of endophytic fungi from *E. breviscapus* roots increased with seasonal fluctuations (Figure 4a).

The beta diversity of *E. breviscapus* fungal communities was evaluated to identify the main drivers of microbial composition. Principal coordinate analyses (PCoAs) were used to compare sample dissimilarities. They were performed based on Unweighted UniFrac distance at the OTU level (Figure 5). As shown in Figure 5a, no clear clustering was found among all samples at the tissue level. In order to further differentiate the variables contributing to the distribution of the endophytic fungal community of *E. breviscapus*, we evaluated the impact of plant tissues and season, separately (Figure 5b–d). In January, samples from leaves and stems tended to cluster together and be distinctly separated from root samples (Figure 5b), while in July, samples from flowers and leaves were grouped together (Figure 5c). Consistent with January, the root samples clustered were relatively isolated (Figure 5b,c). The results suggested that the fungal communities of the root samples exhibited a substantial structural difference compared to other tissues. Regardless of plant tissues, samples were grouped into markedly distinct clusters according to collection times, indicating that season factors may be responsible for shaping the community composition of *E. breviscapus* (Figure 5d).

These results were supported by permutational ANOVA analysis. The test revealed that both plant tissues (R^2^ = 0.3869, *p* < 0.05) and seasons (R^2^ = 0.1945, *p* < 0.05) were important factors contributing to the variation in *E. breviscapus* fungal communities (Table 1).

### 3.4. Biomarkers in Different Organs

To further clarify the biological indicator species that can represent the microbial community characteristics of different *E. breviscapus* tissues, LEfSe analysis [33] was used to analyze the differences in the abundance of endophytes in four tissues. The results showed that the abundance of biomarkers varies in different seasons (Figure 6, Appendix A, LDA score > 4.0).

In January (Figure 6a, Appendix A), at the genus level, the LDA effect size of *Cystofilobasidium*, *Cladosporium*, *Stemphylium*, and *Epicoccum* was larger in flowers. Notably, in leaf samples, the largest LDA effect size was the distribution of the genus *Didymella*. In root samples, *Plectosphaerella*, *Fusarium*, and *Schizothecium* accounted for the greatest percentage of significant differences. In stem samples, *Rhodotorula* and *Filobasidium* were the dominant genera.

In July (Figure 6b, Appendix A), *Cladosporium* remained to be the dominant genus of flower samples. Other dominant genera were *Didymella*, *Filobasidium*, and *Alternaria*, while *Plectosphaerella* was found to be the most dominant genus in leaf samples. In root samples, *Fusarium* also showed a dominant presence during this season. Nevertheless, we did not observe any biomarker genera in stem samples during this month.

### 3.5. Fungal Ecological Functions

FUNGuild was used to predict fungal ecological functions based on community composition. As shown in Figure 7, more than seven trophic modes were classified, including pathotroph, pathotroph–saprotroph, pathotroph–saprotroph–symbiotroph, pathotroph–symbiotroph, saprotroph, saprotroph–symbiotroph, symbiotroph, and unassigned. Moreover, the variation in the relative abundance of fungal function groups was observed in different organs of *E. breviscapus*. The unassigned mode had higher proportions in all tissues (root 36.86% > stem 23.51% > flower 23.42% > leaf 19.77%). In addition, pathotroph–saprotroph was also found to be the dominant fungal function across all samples, with relative abundance greater than 10% (leaf 48.23% > stem31.74% > flower 28.44% > root 17.72%). Interestingly, pathotroph–saprotroph–symbiotroph showed the highest relative abundance (14.01%) in roots and the lowest in leaves (1.93%). Conversely, saprotroph exhibited the highest relative abundance (32.66%) in flowers and the lowest in roots (3.77%).

## 4. Discussion

In this study, fungal endophytic communities associated with *E. breviscapus*, a famous Chinese medicinal plant, were investigated for the first time by high-throughput sequencing. A total of 811 distinct OTUs were generated from *E. breviscapus* at a 97% similarity level. At the phylum level, 10 fungal phyla were recovered by a culture-independent technique, while in a previous study, three phyla (*Ascomycota*, *Deuteromycotina*, and *Mucoromycota*) were isolated from the whole herb of *E. breviscapus* by a culture-dependent method [36]. *Ascomycota* was found to be the most common fungal phylum by both culture-independent and culture-dependent methods. These results were consistent with previous findings that *Ascomycota* was the most frequently encountered fungi in many plant tissues from various environments [37,38]. At the genus level, our analyses suggested that the endophytic community of *E. breviscapus* was predominantly composed of *Didymella*, *Plectosphaerella*, *Filobasidium*, and *Cystofilobasidium*. However, these findings are not entirely congruent with the results obtained by the culture-dependent method, in which the most dominant endophytic fungi were *Alternaria* and *Fusarium*. In the current study, *Alternaria* and *Fusarium* were detected in all samples. Nevertheless, their relative abundance was less than 10.0%. The variation observed can be explained in part by the fact that only a limited number of endophytic microbes were culturable. In contrast, most fungal species were identified by culture-independent techniques compared with the culture method (5 orders, 7 families, 22 genera). Consequently, the proportion of the *Alternaria* and *Fusarium* in total endophytic communities tended to be reduced in the culture-independent study.

Fungal endophytic communities were probably shaped by various environmental and host-related factors, including host species, plant tissue, soil type, geographic location, and climatic effects [39,40,41,42]. In this study, our results revealed that plant tissues and seasons may play a significant role in driving fungal communities of *E. breviscapus*. Only 12.95% of the OTUs were shared among the organs. Moreover, the unique endophyte OTUs of roots were higher than that of other tissues. PCoA plots also showed that the root samples were grouped together and clustered separately compared to the other tissue samples (Figure 5a–c). These results indicate that the endophytic fungi community assembly of roots and other tissues associated with *E. breviscapus* was significantly different. Consistent with these findings, recent studies have demonstrated differences in the above-ground and underground organs of endophytic fungi in some grassland and desert environments [37,38,43]. The variation in microbial communities between the above- and underground may be a consequence of plant selection and the different environments (humidity, temperature, and light intensity) surrounding these tissue niches. Interestingly, when focusing on the proportions of the genera in fungal communities, significant differences could be observed between different plant tissues, as suggested by the results from an analysis of fungal community compositions (Figure 3b) and LEfSe analysis (Figure 6). In January, leaf tissues contained higher numbers of *Didymella* than other tissues, while *Filobasidium* was more abundant in flower and stem tissues than in root and leaf tissues. In root tissues, *Plectosphaerella* was the most dominant genus. By contrast, in July, the dominant genus in the leaves was *Plectosphaerella*. *Filobasidium* represented the dominant genus in both flower and stem samples. In root tissue, unclassified fungi had a major presence (Figure 3b), followed by *Plectosphaerella* and *Fusarium*. These findings indicate that the endophytic assemblages associated with *E. breviscapus* exhibited a certain degree of tissue specificity. The colonization specificity or affinity of fungal endophytes for various plant tissues or organs can be attributed to their ability to utilize specific substrates, including nutrient contents and chemical components to fulfill their nutritional requirements [44]. This specificity is further influenced by host genetics, the microenvironment, and evolutionary selection pressures [44,45]. Similar to the tissue specificity observed in endophytic fungal colonization, the distribution of pharmacologically active compounds in *E. breviscapus* also shows a tissue-specific preference [46,47]. Scutellarin is most concentrated in leaves, followed by stems, flowers, and roots. Conversely, chlorogenic acid and 3,5-dicaffeoylquinic acid are primarily found in roots, with lower levels in stems [46,47]. Is there a potential intrinsic relationship between the dominant endophytic fungal communities within specific plant tissues and the characteristic metabolites produced by those host tissues? Recent studies have explored the associations between endophytes and the primary active compounds in medicinal plants. For instance, Pang et al. (2022) [48] found that endophytic fungal strains *Cyphellophora*, *Sporobolomyces*, and Trichomeriaceae_unclassified were significantly positively correlated with Hup A content (a promising therapeutic candidate for Alzheimer’s disease) in *Huperzia serrata*. Similarly, Ma et al. (2023) [49] demonstrated that *Diaporthe* in the rhizome is significantly positively correlated with the content of key secondary metabolites such as epicatechin in *Fagopyrum dibotrys*. Comparable findings have been reported in other medicinal plants, such as *Rheum palmatum* [50]. Nonetheless, further research is necessary to elucidate the intrinsic relationships between endophytic fungi and key metabolites, particularly within the context of plant tissue specificity.

Alpha diversity analysis indicated that the diversity and richness of the endophytic fungal community in the roots of *E. breviscapus* were the highest in all tissues (Figure 4). Similarly, Wang et al. [44] analyzed the diversity of flora communities in different organs of *Sophora alopecuroides*, an important medicinal plant, and found that the roots had the highest fungal richness and diversity. This may be related to the fact that, in general, soil is considered as one of the prime sources of microbial species diversity and richness [51]. Moreover, there are more rhizosphere exudates and litter residues (shed leaves, broken branches) at the interface between soils and roots. They can provide more hosts and rich nutritional conditions for infection by endophytic fungi, which contribute to the high degree of diversity and richness of fungal communities in roots [44,52].

When focusing on seasonal variations, we found that the alpha diversity indexes of endophytic fungi in samples from July were generally higher than that in samples from January in the same plant tissue, suggesting that seasonal factors may be responsible for increasing local microbial diversity. This may be explained by the fact that in the months (such as July) with sufficient sunshine and suitable temperatures, plants can synthesize more nutrients. These changes were beneficial to the growth and reproduction of endophytic fungi.

FunGuild focuses on fungal trophic type and nutrient management rather than taxonomic identity [53,54,55]. It has been applied to predict the fungal ecological functions in terrestrial and aquatic ecosystems [56,57]. In the present study, we found that the dominant fungal functions in *E. breviscapus* were unassigned, followed by pathotroph–saprotroph. This may be explained by the fact that the FUNGuild database is currently incomplete. Thus, a large number of OTUs were placed in the unassigned group [58]. Additionally, the higher proportion of pathotroph and saprophytic fungi typically indicate that the plants are more susceptible to infestation by pathogenic fungi and prone to a variety of plant diseases [59]. This notion is further confirmed by a recent study. Li et al. [60]. compared and analyzed endosphere fungal communities in healthy and diseased faba bean plants and found that the proportion of pathotroph–saprotroph mode was higher in diseased samples than that in healthy samples. In fact, *E. breviscapus* seem to be highly susceptible to a wide variety of diseases (root rot, leaf spot, and *verticillium wilt*) [3]. This may be closely related to the trophic mode of fungal communities.

It is worth noting that we found some valuable endophytic fungi in *E. breviscapus*. Across all samples, *Didymella* was the most dominant genus present and showed a major presence in leaf samples. Here, *Didymella* has been identified as *Didymella bellidis*. This species has repeatedly been reported as a phytopathogenic fungus that causes leaf spot on *Bellis perennis*, *Angelica gigas*, and *Camellia sinensis* [61,62,63]. These findings suggest that *Didymella bellidis* may be a potential pathogen of leaf spot on *E. breviscapus*. Nevertheless, in terms of the *E. breviscapus* leaf spot pathogen, insufficient information is available. At present, it is generally accepted that leaf spot in *E. breviscapus* is mainly caused by *Alternaria alternate* [64]. Our findings provided a new clue for the thorough investigation of the causative agent of leaf spot on *E. breviscapus*.

Currently, endophytic fungi have been acknowledged as an inexhaustible reservoir of bioactive compounds for drug discovery [65,66]. Over the past few years, numerous promising functional endophytes that produce target metabolites have been found [67,68,69]. It is well established that flavonoids are one of the main components of *E. breviscapus* [70]. We found that some interesting fungi may be involved in the metabolism of flavonoids. In the present study, we observed that flower and stem samples were enriched for *Filobasidium*. Su [71] et al. reported that nobiletin can be metabolized by *Filobasidium magnum* and become bioactive flavonoid derivatives. Their results suggest that *Filobasidium* might promote host plant flavonoid synthesis. Moreover, previous studies confirmed that *Alternaria* sp. and *Fusarium* sp. isolated from *E. breviscapus* were able to produce flavone compounds [72]. Here, these fungi were consistently identified across all collected samples. These findings encourage further investigation into the potential ability of these fungi to produce flavonoids in the future.

## Figures and Tables

**Figure 1 microorganisms-13-01080-f001:**
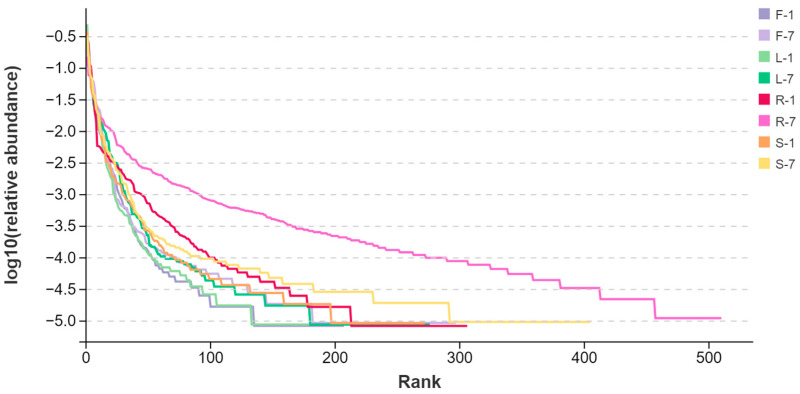
Rank–abundance curve of endophytic fungal communities in *E. breviscapus* based on ITS1 sequences. F, L, R, and S represent flowers, leaves, roots, and stems, respectively. Numbers 1 and 7 stand for the sampling times in January and July. Each group consists of 3 replicates.

**Figure 2 microorganisms-13-01080-f002:**
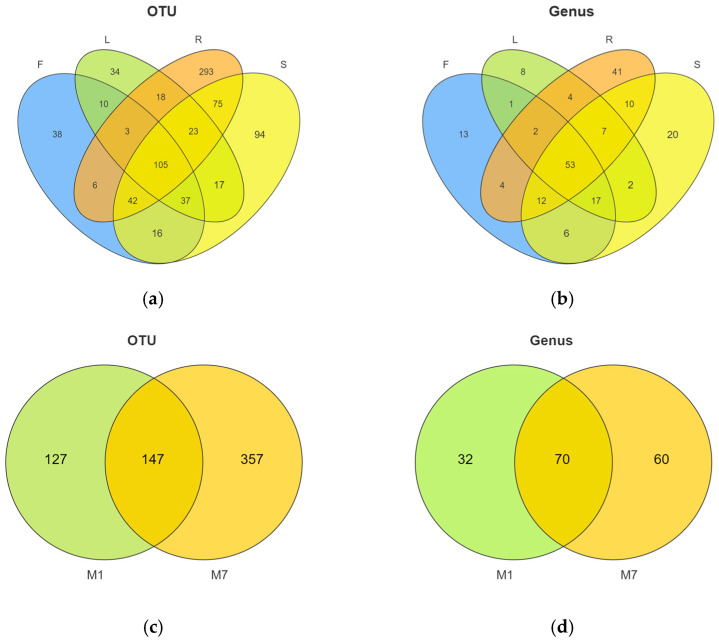
Venn diagram describing the fungal composition among the different organs and seasons. (**a**) OTUs and (**b**) fungal genera distribution difference in endophytic fungi in four organs from *E. breviscapus*. F, L, R, and S represent flowers, leaves, roots, and stems, respectively. (**c**) OTUs and (**d**) fungal genera distribution across *E. breviscapus* in January and July. M denotes the sampling month; 1 and 7 stand for the sampling times in January and July.

**Figure 3 microorganisms-13-01080-f003:**
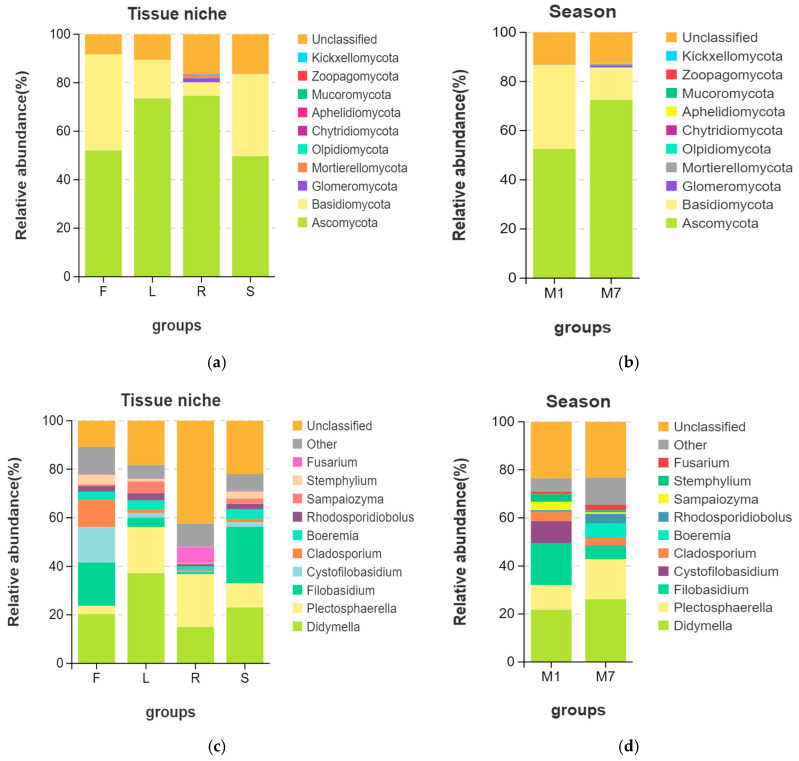
Endophytic fungal composition and relative abundance of *E. breviscapus*. Different tissue niches’ community composition of *E. breviscapus* at the (**a**) phylum level and (**c**) genus level. The fungal composition among the different seasons at the (**b**) phylum level and (**d**) genus level. The box plot shows the top 10 detected species. Other species that can be classified were all assigned to “Other”. F, L, R, and S represent flowers, leaves, roots, and stems, respectively. M denotes the sampling month; 1 and 7 stand for the sampling times in January and July.

**Figure 4 microorganisms-13-01080-f004:**
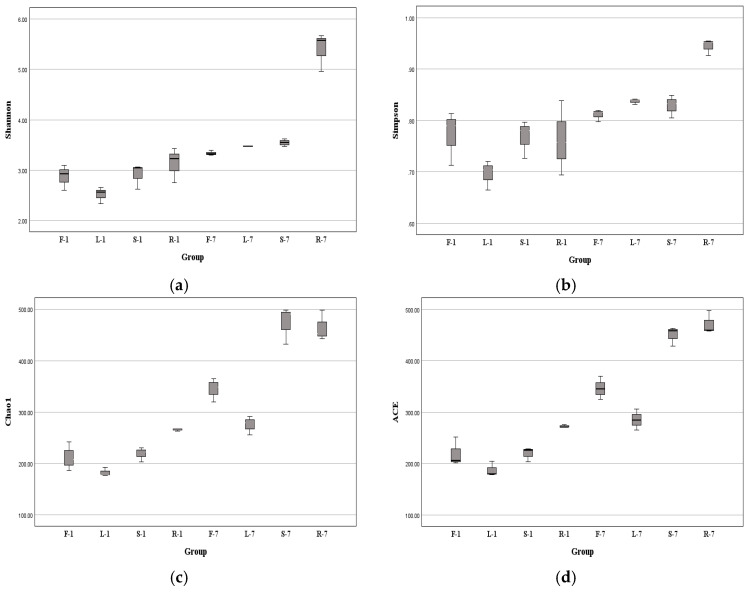
Alpha diversity indexes of endophytic fungi from *E. breviscapus* at different collection times demonstrated by box plots. (**a**) Shannon; (**b**) Simpson; (**c**) Chao1; (**d**) ACE indexes. F, L, R, and S represent flowers, leaves, roots, and stems, respectively; 1 and 7 stand for the sampling times in January and July. Each group consists of 3 replicates.

**Figure 5 microorganisms-13-01080-f005:**
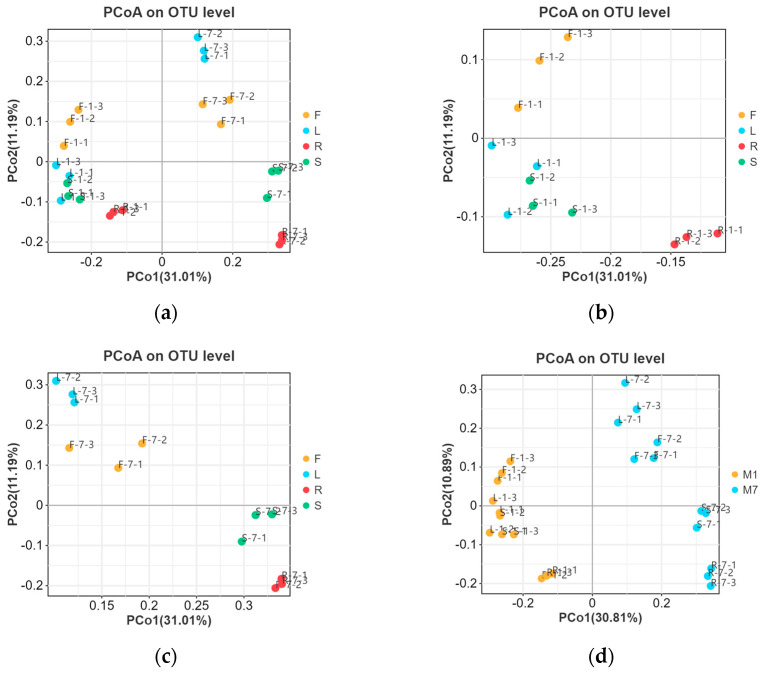
Main factors driving the endophytic microbiota composition of *E. breviscapus*. Principal coordinate analysis (PCoA) based on Unweighted UniFrac distance. (**a**) Plant tissue factor (all 24 samples); (**b**) plant tissue factor (samples were collected in January); (**c**) plant tissue factor (samples were collected in July); (**d**) season factor (all 24 samples). F, L, R, and S represent flowers, leaves, roots, and stems, respectively; M denotes the sampling month; 1 and 7 stand for the sampling times in January and July.

**Figure 6 microorganisms-13-01080-f006:**
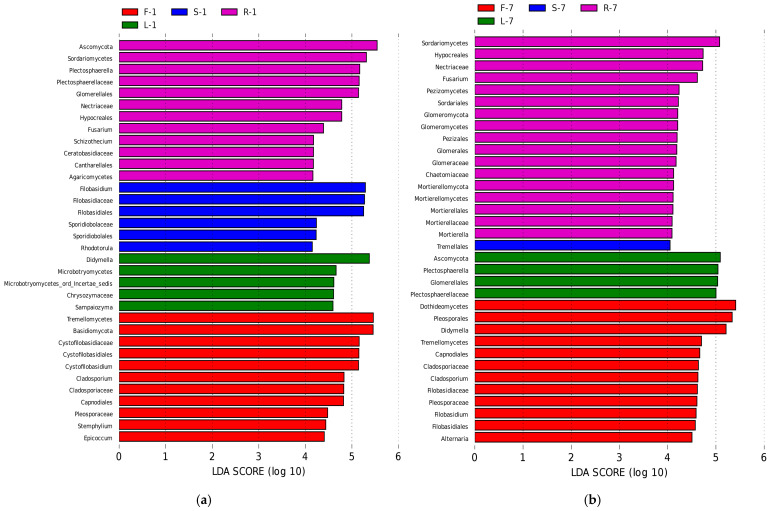
Biomarkers in different organs of *E. breviscapus*. Linear discriminant analysis (LDA) effect size (LEfSe) histograms showing the most differential taxa among four organs of *E. breviscapus*. Circle sizes in the cladogram plot are proportional to fungal abundance. (**a**) Samples were collected in January and analyzed by LEfSe. (**b**) Samples were collected in July and analyzed by LEfSe. All listed taxa were significantly enriched in their respective groups (Kruskal–Wallis test, *p* < 0.05, LDA score > 4). F, L, R, and S represent flowers, leaves, roots, and stems, respectively; 1 and 7 stand for the sampling times in January and July.

**Figure 7 microorganisms-13-01080-f007:**
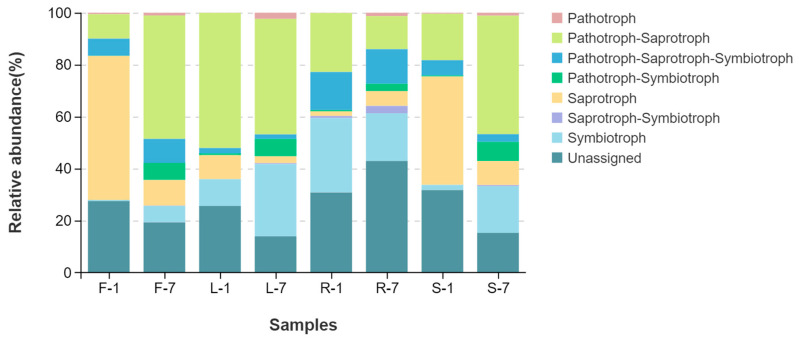
Predicted trophic mode of endophytic fungal communities in various organs of *E. breviscapus*. F, L, R, and S represent flowers, leaves, roots, and stems, respectively; 1 and 7 stand for the sampling times in January and July. Each group consists of 3 replicates.

**Table 1 microorganisms-13-01080-t001:** Effect of tissue and season on fungal communities of *E. breviscapus* revealed using PERMANOVA analysis.

Factor	Df	Sums of Sqs.	Mean Sqs.	F	R^2^	*p*
Tissue	3	1.6996	0.5665	4.2071	0.3869	0.002 *
season	1	0.8545	0.8545	5.3128	0.1945	0.001 *

* *p* < 0.05.

## Data Availability

The data that support the findings of this study are available from the corresponding author upon reasonable request.

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
