# Peer review of "Diversity, Composition, and Ecological Function of Endophytic Fungal Communities Associated with Erigeron breviscapus in China"

_microorganisms, 2025, doi:10.3390/microorganisms13051080_

Round 1

Reviewer 1 Report

Comments and Suggestions for Authors

An interesting article on the study of the diversity of endophytic fungi in different tissues of the Erigeron breviscapus plant in two seasons using high-throughput sequencing and the FUNGuild tool.

The study of fungal endophytes in medicinal plants, especially in the context of their potential role in the synthesis of active compounds, is a research direction consistent with current trends in biotechnology and pharmacognosy. The use of high-throughput sequencing and the FUNGuild tool is a modern approach to the study of microbial communities. PERMANOVA and PCoA are appropriate methods for analyzing the influence of factors on the composition of the community.

The authors indicated the dominant genera of endophytic fungi and their differentiation between tissues. These results can be a basis for further work on the isolation and study of the functions of these microorganisms.

The observation of a greater diversity of fungi in roots in summer suggests an interesting seasonal relationship.

The materials and methods are described correctly.

The authors used an extensive and multifaceted set of statistical and bioinformatic analyses. Both classical alpha- and beta-diversity analysis and assessment of the functional potential of the community were included.

It would be worthwhile to provide more details on the parameters of the tools used and the software versions. The use of current versions of the tools (e.g. QIIME2) could additionally increase the technical value of the work.

Figures are legible. References contain 64 literature items.

Author Response

Response to Reviewer 1 Comments

1. Summary

2. Questions for General Evaluation

Reviewer’s Evaluation

Response and Revisions

Does the introduction provide sufficient background and include all relevant references?

Yes/Can be improved/Must be improved/Not applicable

We thank the reviewer for the positive evaluation of our work and for appreciating the novelty of our study.

Are all the cited references relevant to the research?

Yes/Can be improved/Must be improved/Not applicable

Is the research design appropriate?

Yes/Can be improved/Must be improved/Not applicable

Are the methods adequately described?

Yes/Can be improved/Must be improved/Not applicable

Are the results clearly presented?

Yes/Can be improved/Must be improved/Not applicable

Are the conclusions supported by the results?

Yes/Can be improved/Must be improved/Not applicable

3. Point-by-point response to Comments and Suggestions for Authors

Comments 1: An interesting article on the study of the diversity of endophytic fungi in different tissues of the Erigeron breviscapus plant in two seasons using high- throughput sequencing and the FUNGuild tool. The study of fungal endophytes in medicinal plants, especially in the context of their potential role in the synthesis of active compounds, is a research direction consistent with current trends in biotechnology and pharmacognosy. The use of high-throughput sequencing and the FUNGuild tool is a modern approach to the study of microbial communities. PERMANOVA and PCoA are appropriate methods for analyzing the influence of factors on the composition of the community. The authors indicated the dominant genera of endophytic fungi and their differentiation between tissues. These results can be a basis for further work on the isolation and study of the functions of these microorganisms. The observation of a greater diversity of fungi in roots in summer suggests an interesting seasonal relationship. The materials and methods are described correctly. The authors used an extensive and multifaceted set of statistical and bioinformatic analyses. Both classical alpha- and beta-diversity analysis and assessment of the functional potential of the community were included.

Response 1: We sincerely thank the reviewer for appreciating what this paper shows.

Comments 2: It would be worthwhile to provide more details on the parameters of the tools used and the software versions. The use of current versions of the tools (e.g. QIIME2) could additionally increase the technical value of the work.

Response 2: We agree with the point of the reviewer. These parameters can greatly improve the quality of this manuscript. We have supplemented this information on page 3 (line 124-125, line 131-132) and page 4 (line 135-138) in the revised MS.

Comments 3: Figures are legible. References contain 64 literature items.

Response 3: We thank the reviewer for appreciating the manuscript.

4. Response to Comments on the Quality of English Language

Point 1:

Response 1:  We appreciate the reviewers' constructive feedback on improving the linguistic quality of our manuscript.

5. Additional clarifications

We would like to clarify that owing to budget constraints, the article processing charges (APC) for this manuscript will be supported by an alternative funding source. The relevant grant information has been added to the Funding section of the manuscript.

Reviewer 2 Report

Comments and Suggestions for Authors

The manuscript `Diversity, Composition and ecological function of endophytic fungal communities associated with Erigeron breviscapus in China` presents scientific relevance, as it shows the study on endophytic fungi associated with E. breviscapus, an important Chinese medicinal plant. According to the authors analysis, the study described in the manuscript indicated that tissue and season were the main factors that contribute to the difference in E. breviscapus fungal communities. The authors used high-throughput sequencing technology to examine fungal endophytic community of E. breviscapus.

The manuscript is well written and it should be accepted.

I suggest the authors check English writing as there are some few words spelled incorrectly.

Author Response

Response to Reviewer 2 Comments

1. Summary

2. Questions for General Evaluation

Reviewer’s Evaluation

Response and Revisions

Does the introduction provide sufficient background and include all relevant references?

Yes/Can be improved/Must be improved/Not applicable

We thank the reviewer for the positive evaluation of our work and for appreciating the novelty of our study.

Are all the cited references relevant to the research?

Yes/Can be improved/Must be improved/Not applicable

Is the research design appropriate?

Yes/Can be improved/Must be improved/Not applicable

Are the methods adequately described?

Yes/Can be improved/Must be improved/Not applicable

Are the results clearly presented?

Yes/Can be improved/Must be improved/Not applicable

Are the conclusions supported by the results?

Yes/Can be improved/Must be improved/Not applicable

3. Point-by-point response to Comments and Suggestions for Authors

Comments 1: The manuscript `Diversity, Composition and ecological function of endophytic fungal communities associated with Erigeron breviscapus in China` presents scientific relevance, as it shows the study on endophytic fungi associated with E. breviscapus, an important Chinese medicinal plant. According to the authors analysis, the study described in the manuscript indicated that tissue and season were the main factors that contribute to the difference in E. breviscapus fungal communities. The authors used high-throughput sequencing technology to examine fungal endophytic community of E. breviscapus. The manuscript is well written and it should be accepted.

Response 1: Thank you for appreciating the value of the current work.

Comments 2: I suggest the authors check English writing as there are some few words spelled incorrectly.

Response 2: Sorry, that was an oversight on our part. We have corrected the mistake on page 2, line 65 in the revised MS. Moreover, we have carefully checked the whole text and corrected grammar errors in the revised manuscript.

4. Response to Comments on the Quality of English Language

Point 1:

Response 1:  We appreciate the reviewers' constructive feedback on improving the linguistic quality of our manuscript.

5. Additional clarifications

We would like to clarify that owing to budget constraints, the article processing charges (APC) for this manuscript will be supported by an alternative funding source. The relevant grant information has been added to the Funding section of the manuscript.